# The Effects of 10-Week Strength Training in the Winter on Brown-like Adipose Tissue Vascular Density

**DOI:** 10.3390/ijerph191610375

**Published:** 2022-08-20

**Authors:** Riki Tanaka, Sayuri Fuse-Hamaoka, Miyuki Kuroiwa, Yuko Kurosawa, Tasuki Endo, Ryotaro Kime, Takeshi Yoneshiro, Takafumi Hamaoka

**Affiliations:** 1Department of Sports Medicine for Health Promotion, Tokyo Medical University, Tokyo 160-8402, Japan; 2Faculty of Science and Technology, Meijo University, Nagoya 468-8502, Japan; 3Research Center for Advanced Science and Technology, University of Tokyo, Tokyo 153-8904, Japan

**Keywords:** brown adipose tissue, exercise, resistance training, near-infrared spectroscopy

## Abstract

There is no evidence of the effect of exercise training on human brown-like adipose tissue vascular density (BAT-d). Here, we report whether whole-body strength training (ST) in a cold environment increased BAT-d. The participants were 18 men aged 20–31 years. They were randomly assigned to two groups: one that performed ST twice a week at 75% intensity of one-repetition maximum for 10 weeks during winter (EX; *n* = 9) and a control group that did not perform ST (CT; *n* = 9). The total hemoglobin concentration in the supraclavicular region determined by time-resolved near-infrared spectroscopy was used as a parameter of BAT-d. ST volume (T_vol_) was defined as the mean of the weight × repetition × sets of seven training movements. The number of occasions where the room temperature was lower than the median (NR_cold_) was counted as an index of potential cold exposure during ST. There was no significant between-group difference in BAT-d. Multiple regression analysis using body mass index, body fat percentage, NR_cold,_ and T_vol_ as independent variables revealed that NR_cold_ and T_vol_ were determined as predictive of changes in BAT-d. An appropriate combination of ST with cold environments could be an effective strategy for modulating BAT.

## 1. Introduction

The number of patients with cardiometabolic diseases has increased dramatically worldwide over the past 50 years, partly due to daily energy imbalance [1]. Therefore, there is an urgent need to develop integrated strategies to prevent cardiometabolic disorders and their complications [2]. White adipose tissue (WAT) is used for energy storage. Contrarily, classical brown adipose tissue (cBAT) and beige adipose tissue, a brown-like adipose tissue (BAT) habitable in WAT, are “adaptive heaters” that use excess energy for responding to stimuli such as cold [3], certain drugs, and dietary components with certain modifications by circadian rhythms, exercise training, and aging [3,4,5,6,7]. Activation of BAT improves systemic energy expenditure and glucose and lipid metabolism [8,9,10]. A large cross-sectional study has recently reported that individuals with high BAT activity have a lower risk of developing various non-communicable diseases such as type 2 diabetes mellitus, dyslipidemia, and coronary artery disease [11].

Exercise training is well-known to improve metabolic status by decreasing fat mass [12], increasing skeletal muscle mass (SMM) [13], and improving insulin sensitivity [13]. In addition, previous animal studies have shown the potential of exercise training to increase cBAT and BAT by increasing sympathetic innervation of WAT and endocrine secretion of catecholamines, irisin, and others [14,15,16,17,18,19]. Although very few human studies have been conducted, cross-sectional studies in humans have reported that men and women with endurance exercise training have significantly lower BAT activities (determined by ^18^F-fluorodeoxyglucose (FDG) uptake in the supraclavicular region) than sedentary men and non-athlete women [20,21]. Lower BAT activation in endurance athletes is associated with an insufficient activation of sympathetic nerve activity (SNA) during exercise, which is the major drive to enhance cBAT and BAT activities [22,23]. Further, several previous studies have not provided relevant data on BAT or daily physical activity before and after exercise training [24,25] or data on environmental and seasonal fluctuations in temperature without setting a control group. In contrast to endurance exercise training, strength training is known to cause an explosive increase in SNA outflow, leading to BAT activation [26,27]; however, this has not been focused upon in any previous study and warrants further research investigation.

This study used time-resolved near-infrared spectroscopy (TR-NIRS) to estimate human BAT characteristics by measuring the total hemoglobin concentration ([Total-Hb]) in the supraclavicular region. RNA sequencing data analysis revealed that adult human BAT in the supraclavicular region exhibits a molecular signature similar to beige adipose tissue [28]. Taking advantage of the abundant capillaries in BAT morphology [29,30,31], TR-NIRS detects BAT vascular density (BAT-d) and can safely and non-invasively evaluate BAT-d without requiring cold or radiation exposure [32,33]. A previous study [Total-Hb] assessed by TR-NIRS and standardized uptake values (SUV_mean_) assessed by ^18^F-FDG- positron emission tomography/computed tomography (PET/CT) showed a positive correlation in the supraclavicular region, where BAT was potentially localized [33].

Here, we hypothesized that whole-body strength training in the winter would increase BAT-d. Thus, we aimed to study whether strength training in a cold environment would increase BAT-d and to identify factors related to its changes.

## 2. Materials and Methods

### 2.1. Study Design

On the participant’s first laboratory visit, baseline measurements were conducted to determine body composition, BAT-d, and muscular strength. Within 1 week after the first visit, the exercise group (EX) began exercise training for 10 weeks, twice per week on non-consecutive days for 60 min each at 75% intensity of one-repetition maximum of seven whole-body movements. Exercise training intensity was increased progressively and individually by modulating the weight to be lifted by all participants in the EX once every 2 weeks during the 10-week exercise training program. All exercise training programs were supervised by skilled instructors at all times to ensure that the appropriate weight was handled, adequate rest was taken, the correct form was used for lifting, and risks were managed to prevent injuries and accidents. In all exercise training sessions, the ratio of participants to instructors was 1:1. All measurements were repeated within a week from two days after the end of the exercise training program. While the temperature of the training room was controlled by an air conditioner set at 25.0 °C, the temperature fluctuated from 15.0 to 22.6 °C during training, as the room was relatively large. Then, the relationship between fluctuation in room temperature during training and changes in BAT-d was analyzed as below.

### 2.2. Participants and Interventions Inclusion/Exclusion Criteria

This study’s inclusion criteria were as follows: men aged 20–59 years with a body mass index (BMI) of less than 30 kg/m^2^. Women were not included in the study because the hormonal imbalance caused by the menstrual cycle may affect the whole-body metabolism, including BAT. People with one or more of the following conditions were excluded from the study: daily strength training, smoking, glucose intolerance, dyslipidemia, hypertension, hyperuricemia, gout, coronary artery disease, brain infarction, non-alcoholic fatty liver disease, menstrual disorder, sleep apnea syndrome, motor disorder, obesity related-glomerulopathy, and cardiovascular disease risk. As a result, out of the 19 people who applied for participation in the study, 18 healthy men aged 20–31 years who met the inclusion criteria described above were randomly assigned to the EX (*n* = 9) and control groups (CT; *n* = 9) (Figure 1). This study was conducted per the Declaration of Helsinki, and the study design was approved by the Tokyo Medical University Medical Ethics Review Board (approval number: T2019-0028). Written informed consent was obtained from the participants before conducting the study.

### 2.3. Anthropometric Measurements

Body weight, fat mass, body fat percentage (%BF), and SMM were measured using bioelectric impedance (InBody 720 Body Composition Analyzer; InBody Japan, Tokyo, Japan). BMI was calculated as an individual’s weight (kg) divided by the square of their height (m). Visceral fat area was measured in a standing position using the impedance method (Bioelectrical impedance analysis EW-FA90; Panasonic, Osaka, Japan). The thickness of subcutaneous fat in the supraclavicular, deltoid, and abdominal regions was monitored using B-mode ultrasound (Vscan Dual Probe; GE Vingmed Ultrasound AS, Horten, Norway).

### 2.4. Brown-like Adipose Tissue Vascular Density

BAT-d was evaluated by TR-NIRS (Hamamatsu Photonics K.K., Hamamatsu, Japan) according to previous studies [33,34]. The abundance of capillaries in BAT helped distinguish it from other tissues based on the optical properties of the tissue as evaluated by TR-NIRS. The 3 cm probe used in this study allowed light to reach an average depth of 2 cm [35], which is the depth at which BAT is potentially present [36]. [Total-Hb] in the supraclavicular region, an indicator of BAT-d, was calculated as the sum of oxygenated and deoxygenated hemoglobin, i.e., blood volume. As control regions, [Total-Hb] in the deltoid muscle ([Total-Hb]_delt_) and [Total-Hb] in the abdominal subcutaneous fat ([Total-Hb]_sub_) were measured by using TR-NIRS and BAT-d measurements simultaneously. [Total-Hb] in the supraclavicular region was adjusted for the thickness of the subcutaneous adipose tissue layer [37]. TR-NIRS data were extracted every 10 s and averaged over 1 min. The coefficient of variation within an individual on repeated evaluation was 4.9% [33].

### 2.5. Exercise Training

Before the intervention and after 2, 4, 8, and 10 weeks of the intervention, the EX group underwent a muscular strength test on a strength training machine (AXT-225, TuffStuff Fitness International, Chino, CA, USA), which was evaluated by a health fitness programmer. Muscle strength was assessed for the seven training types: lower limb exercises: leg extension and leg curl; upper limb exercises: chest press, lat pulldown, triceps extension, and biceps curl; and a core exercise of crunches excluding the handgrip strength test. One-repetition maximal muscle strength (1 RM) was not measured directly because the subjects were untrained. For the estimation of the 1 RM value, the Wathen equation was used [38]. Strength training volume (T_vol_) was defined as the mean of the weight × the repetition × sets.

Based on this equation, participants lifted the maximum weight they could lift 1–5 times [38]. All participants practiced with lighter weights and tested their muscle strength during the three trials. If they found that they could perform more than five repetitions with a certain weight, they stopped once, rested until their muscles recovered, and then tried again with a heavier weight. If fewer than five repetitions were performed at maximal muscle strength, the exercise was considered complete.
(1)$$1\;RM=\frac{Weight\;lifted\;per\;repetition\;\left(kg\right)}{(48.8+53.8e^{-0.075\cdot number\;of\;repetition})/100}$$

The handgrip strength test was assessed using a Takei 5401 digital Grip-D hand dynamometer (Takei, Niigata, Japan). Participants stood and extended their elbows carefully, without letting their arms touch their bodies. The participants gripped the dynamometer at maximum effort with a grip width almost perpendicular to the second joint of the index finger. The right–left alternation was performed twice, and the better record of each was averaged.

### 2.6. Questionnaires

Total physical activity was assessed using the International Physical Activity Questionnaire (IPAQ; long version) during a representative week. The IPAQ assesses physical activity lasting longer than 10 min. The total amount of physical activity was calculated as physical activity duration (hours) × physical activity intensity (metabolic equivalents).

The brief-type self-administered diet history questionnaire is a questionnaire designed to examine the amount of nutrients habitually consumed from the diet and to obtain information on individuals’ nutrient and food intakes. We assessed the intake of the three macronutrients: protein, fat, and carbohydrates.

### 2.7. Ambient Temperature

Data on average ambient temperatures were obtained from the Japan Meteorological Agency (Japan Meteorological Agency. “Historical Weather Data”, available at: https://www.jma.go.jp/jma/menu/menureport.html accessed 19 April 2022). The room temperature was contentiously measured during the strength training using a radio clock with a thermometer (8RZ140, CITIZEN, Tokyo, Japan). This thermometer could measure from −9.9 to 50 °C with a measurement error of ±1 °C. Then, the number of occasions with room temperature less than the median (NR_cold_) and the number of occasions with outside temperature less than the median (NO_cold_) were counted as an index of potential cold exposure during the strength training. Participants dressed in personal thin summer clothing during exercise training. Time spent outside (T_out_) during the intervention period and area under the curve (AUC_out_; T_out_ × outside temperature of the training day) were calculated based on the IPAQ questionnaire.

### 2.8. Statistical Analysis

The sample size was calculated considering the type 1 and type 2 errors based on the statistical testing of BAT-d, which was one of the primary outcomes. No previous studies have examined the effect of exercise training on BAT-d. Therefore, we calculated the sample size using the G*Power software (Version 3.1; Bonn University, Bonn, Germany), based on our previous work evaluating the changes in BAT-d in healthy adults who ingested capsinoids or placebo capsules for 8 weeks [39]. In this setting, the sample size was calculated at 80% power and 5% significance level. This resulted in a net sample size of eight subjects in each group. Therefore, we decided to enroll nine subjects in each group in case of dropouts.

Interactions in a two-way repeated-measures analysis of variance (ANOVA) were used to evaluate the pre- and post-intervention changes between the two groups. The difference before and after the intervention was calculated as the amount of change (Δ). Pearson’s correlation coefficient (r) was used for the correlation between the two variables; however, only for the correlation coefficient (r_xy·z_) between ΔBAT and T_vol_, a partial correlation analysis was used to account for the effect of SMM. Stepwise multiple linear regression analysis (MLR) was used to evaluate the independent relationships between parameters (*p* < 0.20) with ΔBAT-d. Data are represented as the mean (standard deviation). All statistical analyses were performed using SPSS Statistics 27 or 28 (IBM SPSS Japan, Tokyo, Japan), and the significance level was set at *p* < 0.05. 

## 3. Results

### 3.1. Participant Characteristics and BAT-d

There was no significant difference in the mean values of all pre- and post-intervention indices in the two groups, except for a significant increase in SMM and muscle strength (Table 1 and Table 2). The change in BAT-d appears to be greater in the control group than in the exercise group, probably due to seasonal variation [34]; however, no significant change was observed (Figure 2).

### 3.2. Ambient Temperature

T_out_ and AUC_out_ had no significant between-group differences (Table 3). The median values of outside temperature on the training day and room temperature during strength training were 10.5 and 18.8 °C, respectively (Table 3). In studies where human BAT is activated by cold stimuli, the room temperature is set below 19 °C, where shivering does not occur.

### 3.3. Relationship between BAT-d and the Number of Occasions Lower Than the Median Room Temperature, the Number of Occasions Lower Than the Median outside Temperature, Skeletal Muscle Mass, and Training Volume

ΔBAT-d is positively correlated with NR_cold_ (Table 4). However, no significant correlation between ΔBAT and NO_cold_ was observed (r = 0.312, *p* = 0.414; Table 4). ΔBAT-d was not associated with pre-SMM (r = 0.461, *p* = 0.212). A partial correlation analysis used to account for the effect of pre-SMM revealed that BAT-d was marginally associated with T_vol_ (r_xy·z_ = 0.690, *p* = 0.058). MLR, using BMI, %BF, NR_cold_, and T_vol_ as independent variables, showed that NR_cold_ and T_vol_ were predictors of ΔBAT-d (Y = −53.59 + 2.49_x1_ + 0.03_x2_ (x1: NR_cold_, x2: T_vol,_), F = 11.23, *p* = 0.009). The contribution was 78.9% (r = 0.888) and the standard error of the estimate was 5.22 µM (Table 4). On the other hand, Δ[Total-Hb_delt_] and Δ[Total-Hb_sub_] as parameters of the control region were not correlated with NR_cold_ (r = 0.258, *p* = 0.504 and r = −0.111, *p* = 0.776, respectively) and T_vol_ (r = 0.478, *p* = 0.193 and r = 0.139, *p* = 0.721, respectively).

## 4. Discussion

Contrary to our hypothesis, ΔBAT-d did not significantly differ between the EX and CT groups. MLR using a stepwise method with BMI, %BF, NR_cold_, and T_vol_ as independent variables revealed that NR_cold_ and T_vol_ were factors predicting ΔBAT-d, suggesting that some stimuli such as elevated SNA occurring due to a lower room temperature and strength training volume may be associated with the increase in BAT-d.

We have observed the large interindividual variability in the change in BAT-d following strength training and the same for the control group. The interindividual heterogeneity in BAT-d dynamics pre- and post-intervention may be due to the susceptibility to seasonal ambient temperature fluctuation in CT and/or the stimuli elicited by the strength training in EX. Further research is needed to clarify the interindividual variability and the physiological mechanisms behind these findings. Similar to the present results, one previous study reported that ^18^F-FDG uptake in the supraclavicular region did not change after 2 weeks of high-intensity cycling, which would have led to adequate activation of SNA [40]. One previous study in an animal model reported that high-intensity resistance training, consisting of 50–100% of maximal load, five times a week for 8 consecutive weeks, induced the browning of inguinal and retroperitoneal white adipose tissue or increased BAT [16]. In one animal model study, BAT weights were comparable between the cold acclimation group (CG) and cold acclimation–exercise training group (CTG). However, BAT lipid content was 9.3% lower in the CTG than in the CG [41], which would be a marginal difference in lipid density resulting in a small increase in BAT-d when adopted in our study. In this study, the most important reason for comparable results of BAT-d in the EX and CT was the increase in body temperature due to exercise training that may have offset the increase in BAT-d due to the increased SNA.

In other words, the rapid increase in body temperature associated with increased thermogenesis due to exercise training is similar to a state of intermittent mild heat exposure. As heat exposure has been reported to have an inhibitory effect on norepinephrine (NE)-dependent non-shivering thermogenesis (NST) [42], it is possible that the increase in body temperature due to strength training also inhibited the increase in BAT-d in this study. As BAT is essentially an adaptive thermogenic tissue in cold environments, the attenuated increase in BAT-d as a whole in this study might be a reasonable outcome. The association of strength training volume with ΔBAT-d indicates that a larger amount of muscle work elicits a greater efferent increase in SNA through larger stimuli. In fact, subjects with greater muscle mass or strength easily induce higher intramuscular pressure resulting in spontaneous blood flow restriction, which, in turn, creates enhanced SNA [43,44,45].

Another possible reason for the comparable BAT-d between EX and CT in this study could be that the increase in SNA during strength training was lower than that during cold exposure. In an animal study, the exercise training group (EXG; 90.6 g (SD 11.9)) had a lower BAT weight than the chronic cold exposure group (CCE; 283.7 g (SD 25.2)). Furthermore, the total amount of urinary catecholamine metabolites such as 3-methoxy-4-hydroxymandelic acid (VMA) and homovanillic acid (HVA) was significantly lower in the EXG than in the CCE. On the other hand, urinary catecholamine metabolites per unit time were higher in the EXG (VMA: 15.6 µg/h (SD 0.57), HVA: 9.94 µg/h (SD 0.27)) than in the CCE (VMA: 7.80 µg/h (SD 1.96), HVA: 1.94 µg/h (SD 0.67)). In short, it is suggested that prolonged, low-dose cold exposure may enhance NE-dependent NST better than that with short-duration, high-dose SNA stimulation derived from exercise training [46].

Cold exposure is the primary factor that activates BAT [5]. The present strength training program was selected according to the American College of Sports Medicine exercise guidelines that allow the major muscle groups to be trained twice a week [47]. However, SNA created by the strength training in the present study may be lower than that of a two-hour cold exposure for 6 weeks obtained in a previous study reporting an increase in BAT activity and a decrease in body fat mass in healthy young men [48]. A lower amount of SNA may be one of the reasons for no increase in BAT by the current strength training. However, increasing exercise volume to enhance exercise-induced SNA may suppress BAT because it causes a greater increase in body temperature in humans. Quantitative assessment of SNA during a combination of cold exposure with exercise training should be investigated to determine measures to effectively increase BAT, which would be a more comfortable method than a high-stress cold-exposure regimen itself.

### Limitations

This study has four limitations. Firstly, while TR-NIRS allows noninvasive measurements, it may be less valid than invasive tissue biopsies. BAT-d measurement using TR-NIRS has been validated over and over. In this study, only [Total-Hb] in the supraclavicular region ([Total-Hb]_sup_), not [Total-Hb]_delt_ or [Total-Hb]_sub_, was significantly correlated with T_vol_ and NR_cold_. Furthermore, previous studies have reported that [Total-Hb]_sup_ can be characterized distinctly from [Total-Hb]_sub_ [49]. To further enhance the validity of this study, it is necessary to link observations to tissue biopsy. However, we realize that obtaining this in a human subject study is challenging. Instead, we can examine our hypotheses in part through the use of animal models. Secondly, because blood hemoglobin levels were not measured in this study, it was not possible to determine changes in blood hemoglobin levels before and after exercise and in the control group. Among the limited longitudinal investigations, Schobersberger et al. [50] reported that 6 weeks of strength training slightly increased erythropoiesis. Others have shown that 12 weeks of strength training either has no effect on erythrocyte volume and hemoglobin concentration [51] or decreases hemoglobin concentration with no change in hematocrit [52]. One strength training study showed no significant changes in hematocrit or red blood cell count at 10 weeks, but an increase in hematocrit and red blood cell count was observed at 20 weeks. Further research is needed to clarify the effects of strength training on blood hemoglobin levels and the physiological mechanisms behind these findings. Thirdly, the room temperature during strength training was not controlled due to the relatively large room size and the lack of an air conditioner to effectively control the temperature. However, coincidently occurring temperature variations allowed us to examine the relationship between the occurrence of cold exposure during strength training and changes in BAT-d. Finally, only young male participants were included. As this is a pilot study investigating the effects of exercise on human BAT, women were not included due to the wide fluctuations in hormones during the menstrual cycle. In the future, it will be necessary to monitor hormonal imbalances during the menstrual cycle and then study sex differences in exercise-related changes in BAT. It is also important to study the effect of exercise on BAT among different sexes by monitoring the hormone balance of the menstrual cycle.

## 5. Conclusions

There were individual differences in ΔBAT-d and no overall increase in BAT-d, possibly due to negative factors such as increased body temperature elicited from strength training. However, when the analysis focused on individual data, the fact that both NR_cold_ and T_vol_ were independently extracted as factors explaining ΔBAT-d in the MLR suggests that an appropriate combination of strength training in a mildly cold environment could be a strategy for effective and comfortable modulation of BAT.

## Figures and Tables

**Figure 1 ijerph-19-10375-f001:**
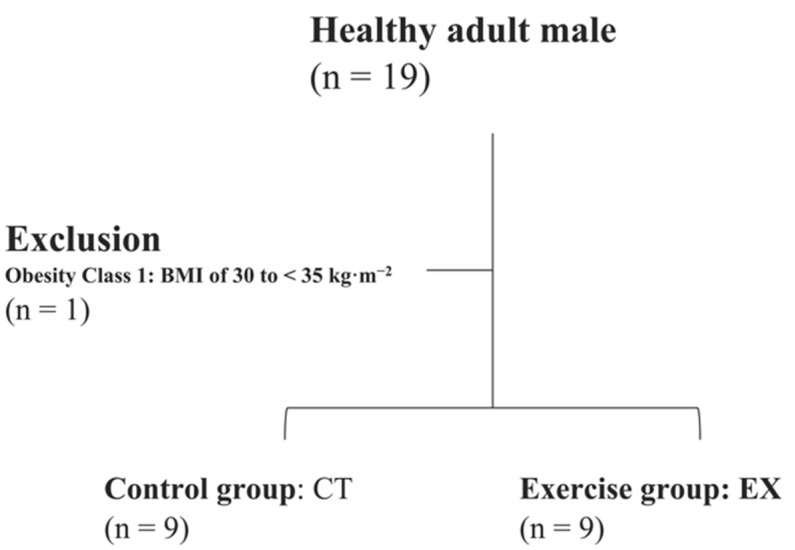
Inclusion criteria.

**Figure 2 ijerph-19-10375-f002:**
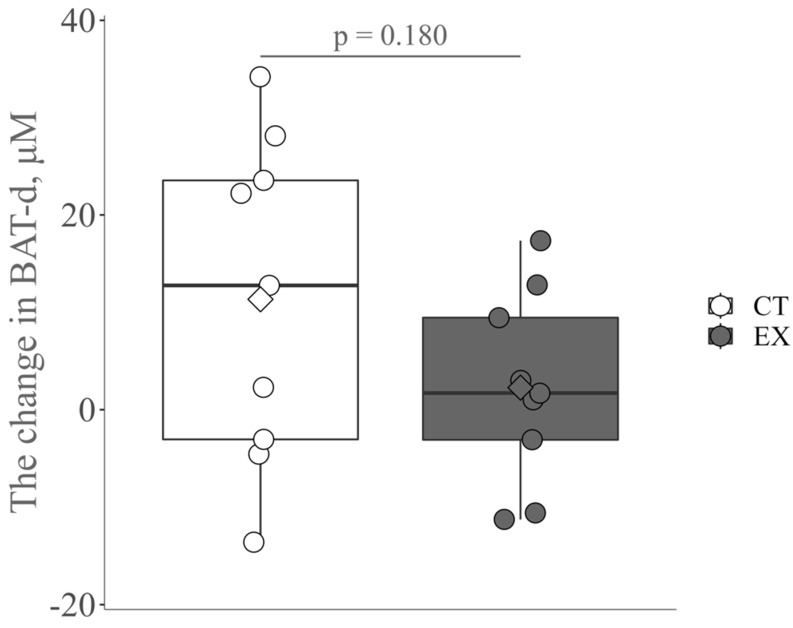
The change in brown-like adipose tissue vascular density (BAT-d) pre- and post-intervention in the control group (CT) and the exercise group (Ex). The circle and diamond plots indicate individual values and the mean values, respectively.

**Table 1 ijerph-19-10375-t001:** Participant body composition.

	CT		EX		Interaction		Main Effects (Group)		Main Effects (Intervention)	
	Pre	Post	Pre	Post	*p*	Partial Eta Square (η^2^_p_)	*p*	Partial Eta Square (η^2^_p_)	*p*	Partial Eta Square (η^2^_p_)
Age (years)	22 (1)	22 (1)	24 (4)	24 (4)	-	-	-	-	-	-
Height (m)	1.72 (0.06)	1.72 (0.06)	1.72 (0.06)	1.72 (0.06)	0.320	0.062	0.853	0.002	0.392	0.046
Body mass index (kg·m^−2^)	23.1 (3.5)	23.3 (3.3)	21.8 (2.9)	22.1 (2.8)	0.746	0.007	0.410	0.043	0.074	0.186
Body fat percentage (%)	19.7 (9.0)	20.0 (9.0)	18.8 (5.6)	17.9 (5.1)	0.109	0.152	0.677	0.011	0.387	0.047
Skeletal muscle mass (kg)	30.7 (4.0)	30.9 (3.8)	29.3 (4.5)	30.2 (4.4)	0.016	0.313	0.581	0.019	<0.001	0.561
Waist circumference (cm)	78.0 (7.7)	79.5 (7.1)	74.9 (8.7)	76.9 (7.1)	0.752	0.006	0.433	0.039	0.056	0.210
Visceral fat area (cm^2^)	39.9 (27.5)	44.2 (30.5)	42.1 (48.4)	35.9 (23.7)	0.264	0.123	0.714	0.014	0.573	0.033
BAT-d (μM)	74.1 (19.7)	85.4 (26.0)	76.4 (28.2)	78.6 (24.7)	0.180	0.109	0.847	0.002	0.052	0.217
[Total-Hb]_delt_ (µM)	101.0 (24.0)	92.5 (19.4)	98.8 (20.7)	98.9 (15.3)	0.184	0.107	0.817	0.003	0.200	0.100
[Total-Hb]_sub_ (µM)	41.2 (33.3)	36.9 (27.2)	32.8 (20.0)	35.7 (20.3)	0.232	0.088	0.691	0.010	0.806	0.004

Mean (standard deviation (SD)). Two-way repeated-measures ANOVA was used to assess the pre- and post-intervention changes between the two groups. BAT-d: total hemoglobin concentration in the supraclavicular region as a parameter of brown-like adipose tissue vascular density; [Total-Hb]_delt_: total hemoglobin concentration in the deltoid muscle as a parameter of control region; [Total-Hb]_sub_: total hemoglobin concentration in the abdominal subcutaneous fat as a parameter of control region.

**Table 2 ijerph-19-10375-t002:** Participants’ exercise training and dietary status.

	CT		EX		Interaction		Main Effects (Group)		Main Effects (Intervention)	
	Pre	Post	Pre	Post	*p*	Partial Eta Square (η^2^_p_)	*p*	Partial Eta Square (η^2^_p_)	*p*	Partial Eta Square (η^2^_p_)
Leg extension (N)	763.4 (280.3)	757.5 (288.1)	812.4 (213.6)	1213.2 (242.1)	<0.001	0.677	0.045	0.228	<0.001	0.663
Chest press (N)	492 (75.5)	492 (69.6)	513.5 (103.9)	694.8 (115.6)	<0.001	0.814	0.067	0.194	<0.001	0.832
Grip strength (N)	365.5 (63.7)	381.2 (52.9)	375.3 (62.7)	384.2 (51.0)	0.549	0.023	0.808	0.004	0.055	0.211
Training volume (N)			12,731.2 (1834.6)							
Protein intake (g·day^−1^)	82.3 (35.9)		94.9 (34.4)							
Fat intake (g·day^−1^)	68.8 (28.6)		74.8 (21.4)							
Carbohydrate intake (g·day^−1^)	306.3 (138.0)		257.4 (94.4)							
Physical activity (METs·h·day^−1^)	6.4 (3.2)		3.8 (3.2)							

Mean (SD). A two-way repeated-measures ANOVA was used to assess the pre- and post-intervention changes between the two groups.

**Table 3 ijerph-19-10375-t003:** Ambient temperature.

	CT		EX		
	Mean	Median	Mean	Median	*p*
T_out_ (h)	347.2 (186.3)	426.0 [146.5–502.5]	280.6 (276.9)	152.0 [84.5–487.0]	0.558
AUC_out_ (h·days)	4194.5 (2525.2)	4016.4 [1754.2–6602.6]	3231.4 (3500.9)	1444.3 [1064.2–4892.2]	0.513
Outside temperature of the training day (°C)	-	-	11.1 (2.0)	10.5 [9.7–12.4]	-
NO_cold_ (times)	-	-	10.3 (3.1)	10.0 [8.5–13.0]	-
Room temperature during strength training (°C)	-	-	18.7 (0.8)	18.8 [ 17.8–19.5]	-
NR_cold_ (times)	-	-	8.4 (2.2)	8.0 [6.5–9.5]	-

Values are means (SD) or medians [first–third quartile]; an unpaired *t*-test was used to compare the two groups.

**Table 4 ijerph-19-10375-t004:** Multiple linear regression analysis for ΔBAT-d.

ΔBAT-d (µM)	Univariate Regression		Multivariate Regression		
	r	*p*	B	β	*p*
Pre_BMI (kg·m^−2^)	0.562	0.116	0.153	-	0.705
Pre_%BF (%)	0.502	0.169	0.182	-	0.514
NR_cold_ (times)	0.754	0.019	2.493	0.554	0.035 *
T_vol_ (N)	0.728	0.026	0.027	0.51	0.046 *
			R^2^ = 0.789, SEE = 5.216		

B: unstandardized beta; β: standardized partial regression coefficient; r: correlation coefficient; SEE: standard error of estimate; R^2^: coefficient of determination; *: *p* < 0.05.

## Data Availability

Derived data supporting the findings of this study are available from the corresponding author T.H. on request.

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
