# Peer review of "The Effects of 10-Week Strength Training in the Winter on Brown-like Adipose Tissue Vascular Density"

_ijerph, 2022, doi:10.3390/ijerph191610375_

Round 1

Reviewer 1 Report

The study examined the effect of whole-body strength training on human beige adipose tissue vascular density (BgAT-d) in a cold environment. The study provided some insight in the field since little study has examined the effect of strength training on human brown/brown-like adipose activity. But the manuscript is yet to be improved before it reaches the publication standard of the journal.

Major points: 

1. The key technique here used is the TR-NIRS-based measurement of BgAT-d. Although the same group has published their study using the same technique, this method is still not widely used. Thus to add to the validity of the current study, TR-NIRS measurement should be performed on other anatomical locations of the people, such as the subcutaneous adipose, skeletal muscle, and etc.

2. How about the change in total hemoglobin concentration in blood before and after exercise and in control group?

3. Current understanding is that the adipose tissue in the supraclavicular region is a mixture of brown and beige fat. Therefore it is not suitable to make the affirmative notion here that the effect observed is restricted to beige adipocytes. “Brown-like adipose tissue” would be more appropriate.

Minor points:

1. Language should be check through since there are a number of typos.

Reviewer 2 Report

This pilot study unveiled that BgAT could not be changed after 10-week of cold-exposed training, However, the study showed that SMM and muscle strength increased after the intervention.

Increment in body temperature could be the supporting reason for the results of the study.  As a result, no significant change had been observed in the study.

The author represents the proper study to limit the effect of participant variations as shown on baseline characteristic data that was decent.

Some study limitations had been mentioned in the last section of the discussion such as regulation of room temperature, and proper explanation of not including female participants in the study. 

In the reference section, please remove another set of references (line no. 429 onward) as it was typo mistakes or not being checked properly before submission.

Round 2

Reviewer 1 Report

The revised version has been improved. 

Author Response

We would like to thank reviewer #1 for the valuable suggestions and comments.

A certificate of proofreading is attached. 
